# Effect of Se on Structure and Electrical Properties of Ge-As-Te Glass

**DOI:** 10.3390/ma15051797

**Published:** 2022-02-27

**Authors:** Kangning Liu, Yan Kang, Haizheng Tao, Xianghua Zhang, Yinsheng Xu

**Affiliations:** 1State Key Laboratory of Silicate Materials for Architectures, Wuhan University of Technology, Wuhan 430070, China; 18637494033@163.com (K.L.); ky19980414@163.com (Y.K.); thz@whut.edu.cn (H.T.); xzhang@univ-rennes1.fr (X.Z.); 2ISCR (Institut Des Sciences Chimiques de Rennes)—UMR 6226, CNRS, Université de Rennes 1, 35042 Rennes, France

**Keywords:** chalcogenide glass, thermal property, optical property, electrical property, Raman spectra

## Abstract

The Ge-As-Te glass has a wide infrared transmission window range of 3–18 μm, but its crystallization tendency is severe due to the metallicity of the Te atom, which limits its development in the mid- and far-infrared fields. In this work, the Se element was introduced to stabilize the Ge-As-Te glass. Some glasses with Δ*T* ≥ 150 °C have excellent thermal stability, indicating these glasses can be prepared in large sizes for industrialization. The Ge-As-Se-Te (GAST) glasses still have a wide infrared transmission window (3–18 μm) and a high linear refractive index (3.2–3.6), indicating that the GAST glass is an ideal material for infrared optics. Raman spectra show that the main structural units for GAST glass are [GeTe_4_] tetrahedra, [AsTe_3_] pyramids, and [GeTe_4_Se_4−*x*_] tetrahedra, and with the decrease of Te content (≤50 mol%), As-As and Ge-Ge homopolar bonds appear in the glass due to the non-stoichiometric ratio. The conductivity *σ* of the studied GAST glasses decreases with the decrease of the Te content. The highest *σ* value of 1.55 × 10^−5^ S/cm is obtained in the glass with a high Te content. The activation energy *E*_a_ of the glass increases with the decrease of the Te content, indicating that the glass with a high Te content is more sensitive to temperature. This work provides a foundation for widening the application of GAST glass materials in the field of infrared optics.

## 1. Introduction

Infrared (IR) technology has become important in the development of modern optics, and its development mainly depends on the research of IR optical materials and detectors. There are many IR materials, such as germanium (Ge), zinc selenide (ZnSe), zinc sulfide (ZnS), and gallium arsenide (GaAs) [1]. However, the high cost and the requirement of the diamond turning for processing limit the application of IR crystal materials in optics. Chalcogenide glasses have excellent transmittance from visible to far IR, good physical and chemical properties, and can use high-precision molding technology to produce lens. Compared with single crystal Ge, it has low long wavelength dispersion and can be complementary with Ge to eliminate chromatic aberration. Moreover, the dn/dT of chalcogenide glass is one tenth of germanium and its transmittance does not vary with temperature.

Chalcogenide glasses can be divided into S-, Se-, and Te-based glass according to the anion composition of the matrix glass. Among them, S- and Se-based alloys are relatively easy to form glass. Many chalcogenide glasses have good thermal stability and are easy to pull into fibers, but their IR transmittance cut-off edges are only 11 μm and 15 μm, respectively [2,3]. Besides this, to eliminate the advanced spherical aberration and enlarge the field of view and aperture angle of the optical apparatus [4], the linear refractive index of the IR lens needs to be increased, which is closely related to the density and ionic polarization of chalcogenide glasses. Therefore, it is necessary to introduce heavy Te elements into the glasses to extend the optical window and improve the refractive index. However, the Te atom has a strong metallic property, and thus cannot form a stable glass state and is easy to be crystallized. It is found that the introduction of electron-deficient elements into the melt to localize the free electrons in the alloy is an effective strategy against the nucleation tendency of Te-based glass microcrystals [5,6]. To obtain stable Te-based glass, researchers have tried a variety of Te-based binary glasses, such as Ge-Te [7] and As-Te [8], which have narrow glass-forming regions and poor thermal stability. Savage et al. [1] first reported the glass-forming region of the Ge-As-Te ternary glass in 1979, but the glass-forming region was narrow and mainly concentrated in the rich Te phase, which was biased towards the As-Te side. Savage et al. [9] added the Te element to Ge-As-Se ternary glass in 1980 to investigate the influence of the Te element introduction on the thermal properties of Ge-As-Se ternary glass and evaluate the application potential of quaternary Ge-As-Se-Te (GAST) glass as an IR optical material in the wavelength range of 8–12 μm. Tikhomirov et al. [10] reported that the optimized GAST glass has a high glass-transition temperature and no crystallization peak on the DSC curve. Its transmission window range is 1.5 to 23 μm, and the refractive index can be changed from 3.2 to 3.4 by varying the ratio of Se/Te or Ge/As, making it an ideal candidate for mid-IR optics.

As a semiconductor glass, Te-based glass usually has high conductivity because of the strong metallic property of tellurium. Hegab et al. studied the AC conductivity and dielectric properties of Ge-Se-Te glass film, and explained the reasons for the change of AC conductivity with temperature and frequency, which provides useful information about the conduction mechanism in the considered film compositions [11]. Yang et al. reported the suitability of the Ge-As-Te system and the Cu-Ge-As-Te system in bio-optical detection due to their high conductivity and high IR transmission characteristics; some excellent compositions have relatively high conductivities (near 10^−4^ S/cm) and a wide IR transmission range (3–18 μm) [12]. Dwivedi et al. reviewed the preparation, optical properties, and characterization techniques of the Se–Te-based glasses and their potential applications in the field of optoelectronics [13].

IR glass with a high refractive index can shorten the optical path for optical design. The Ge-As-Te glass has almost the highest refractive index among chalcogenide glasses; however, Te-based chalcogenide glass suffers from a high crystallization tendency due to the strong metallicity of the Te atom. It was found that the introduction of an appropriate amount of selenium can greatly improve the glass-forming ability and effectively broaden the glass-forming range [14,15]. Hence, the novelty of this work is to study the Se-stabilized Ge-As-Te glass. In this work, we introduced an appropriate amount of Se into Ge-As-Te glasses and studied the thermal stability and the optical and electronic properties of these glasses. This work can provide comprehensive data to design lenses for IR optics.

## 2. Materials and Methods

Two series of samples with a composition of Ge_10_As_90−*x*_Te*_x_* (*x* = 80~30, in mol%) and Ge_20_As_80−*y*_Te*_y_* (*y* = 70~30, in mol%) were selected to introduce different ratios of Se (5, 10, and 14, in mol%). As shown in Table 1, 33 samples were prepared with the appropriate amounts of high purity (6N) Ge, As, Se, and Te elements, respectively. The batch was put into a quartz ampoule (*Φ*10 mm) and sealed in a vacuum (10^−4^ Pa). The ampoule was then placed in a homemade rocking tube furnace and heated at 850 °C for 20 h. Afterward, the melt inside the ampoule was quenched with ice water and annealed at the temperature of 10 °C below the *T*_g_ for 2 h to eliminate the internal stress of the glass. Finally, the glass rod was taken from the quartz tubes, cut into slices (*Φ*10 mm × 2 mm), and then precisely polished for the optical measurement.

Differential scanning calorimetry (DSC; STA449F1, NETZSCH, Berlin, Germany) was used to determine the glass transition temperature *T*_g_ and onset of crystallization temperature *T*_x_. A quantity of 25 mg of small-particle sample sealed in an aluminum crucible was used for test at the temperature range of 25–450 °C. The X-ray powder diffraction patterns of all the samples were collected by an X-ray diffractometer (XRD; D8 Discover, Bruker, Karlsruhe, Germany) in the range of 10–80°. The transmittance of glass samples was measured using Fourier transformed-IR spectrometer (FTIR; INVENIO S, Bruker, Ettlingen, Germany) with a thickness of about 1.5 mm. The measurements of refractive index *n* were conducted using an IR-variable angle spectroscopic ellipsometer (IR-Vase Mark II, J.A. Woollam co, Ltd., Lincoln, NE, USA) which can achieve high-precision measurements with an error of ±0.001. Raman spectra were recorded on Raman spectrometer (LabRam HR Evolution, Horiba Jobin Yvon, Paris, France). The excitation wavelength is 633 nm. The AC electrical conductivity was measured by an electrochemical workstation (CHI600E, Chenhua Instrument Co, Ltd., Shanghai, China) from 293 to 333 K, with a frequency range from 800 kHz to 1 Hz. Samples were coated with gold film on each face and gold wire was connected to the coated sample with silver paste.

## 3. Results and Discussion

### 3.1. Glass Properties

As shown in Figure 1, the glass-forming region of the Ge-As-Te glass is redrawn according to the references [16,17]. Nine compositions in the glass-forming region and two compositions in the boundary of the glass-forming region were selected for further study. After the introduction of the Se element, all the samples show an amorphous state, as confirmed by the XRD patterns with two broad diffuse scattering halos (Figure 2), indicating the introduction of the Se element can increase the glass-forming ability. However, the XRD pattern of the C1 sample with the high Te composition presents sharp diffraction peaks, as shown in Figure 3. The C1 sample has a high Te content. When Se atoms are introduced into the Ge-As-Te glass network structure, due to the non-stoichiometric ratio, Se atoms will preferentially bond with Ge and As atoms, resulting in a Te atom surplus in the original glass structure. Hence, the C1 sample has been crystallized because of the metallic property of the Te atom.

According to the thermal stability parameter proposed by Hruby [8], ∆*T* (defined as ∆*T* = *T*_x_ − *T*_g_) is proportional to the trend of the glass-forming ability; the larger the ∆*T*, the better the thermal stability of the glass. When the ∆*T* is larger than 150 °C, the glass is hard to be crystallized Table 1 shows the DSC results and the physical properties of the GAST glasses. To explore the influence of the Te contents on the *T*_g_ and *T*_x_ of the GAST glass, the Se_5_(Ge_0.1_As_0.9−*x*_Te*_x_*)_95_ glasses were selected.As is shown in Figure 4, when the Se content was fixed, the *T*_g_ of the Se_5_(Ge_0.1_As_0.9−*x*_Te*_x_*)_95_ glasses increased from 124 °C to 194 °C with the decrease of Te content, and no crystallization peak appeared in the glass compositions with a Te content less than 50 mol%, indicating that the thermal stability of the glass becomes better. Generally, the *T*_g_ value is closely related to the network structure of the glass. In GAST glass, As and Ge atoms form [AsX_3_] pyramids and [GeX_4_] tetrahedra (X = Se and Te), respectively, and the [AsSe_3−*x*_Te*_x_*] and [GeSe_4−*x*_Te*_x_*] units can also exist in the glass structure [18]. When Te is replaced by As, which is a glass former, the number of [AsX_3_] or [AsSe_3−*x*_Te*_x_*] will increase obviously, thus the *T*_g_ values of the Se_5_(Ge_0.1_As_0.9−*x*_Te*_x_*)_95_ glasses show an increasing trendency. However, when the ratio of the Ge-As-Te ternary composition is fixed, the *T*_g_ values of the GAST glasses are irregular changes with the increase of the Se content. Since the Se element is added on the basis of the fixed Ge-As-Te ternary compositions in the formula design, the content of Ge and As decreases with the increase of the Se content compared with the original matrix glass. Therefore, it is difficult to measure the number of glass network-forming bodies qualitatively in the glass network structure, resulting in irregular *T*_g_ changes.

In addition, according to the topological constraint theory, the connection degree of the glass network structure is related to the mean coordination number (MCN). When MCN < 2.4, the glass network structure is relatively loose, so the *T*_g_ and Δ*T* values of A1 and B1 glass are low, the thermal stability of the glass is poor, and the C1 sample is even crystallized. With the increase of the MCN value, the glass network structure is more and more closely connected, the corresponding *T*_g_ value of the glass increases, and the thermal stability also increases.

### 3.2. Optical Properties

Figure 5a shows the transmission spectra of B2–B6: Se_10_(Ge_0.1_As_0.9−*x*_Te*_x_*)_90_ (*x* = 0.7 − 0.3) glasses with the range of 1–25 μm. Due to the small ∆*T* of the B1: Se_10_(Ge_0.1_As_0.1_Te_0.8_)_90_ glass, it is difficult to form in bulk and polish into thin slices, so its IR transmission spectrum is not studied. As shown in Figure 5a, with the increase of the Te content, the near-IR cut-off wavelength moves from 1.37 μm to 1.84 μm, and the far-IR cut-off wavelength can reach 25 μm (cut-off edge is defined as the wavelength at half of the maximum transmittance of the glass sample). The IR transmittance of glass is mainly affected by the reflection, scattering, and absorption of the glass. When a beam of light passes through the surface of double-side polished GAST glass, Fresnel loss will occur, and its reflectivity *R* is related to the refractive index *n* of the glass itself. The linear refractive index of the GAST glass sample is about 3.2–3.6 (Figure 6), so the transmittance has about 27–32% Fresnel reflection loss. However, with the increase of Te content, the transmittance of the glass samples decreases significantly, and the transmittance of the B2 sample is only about 40%. In addition, the transmittance of the B2, B3, and B4 glass samples in the long-wave region is higher than that in the short-wave region (with a limit of 10 μm). These differences may be caused by the formation of nano-crystals, which were not identified by XRD [19]. The rapid decrease in transmittance between 18 and 25 μm can be attributed to the mixing of multiple phonons in the glass sample [20]. Besides this, the impurity absorption peak of the GAST samples is mainly concentrated at 13.4 μm, as shown in Figure 5b, which is composed of As-O, Ge-O, and Se-O bonds [21]. The absorption band at 13.4 μm can be effectively eliminated by adding an oxygen-getter, such as Mg foil, and a relatively flat transmission spectrum can be obtained.

The refractive index of A2–A6: Se_5_(Ge_0.1_As_0.9−*x*_Te*_x_*)_95_ (*x* = 0.7 − 0.3) and A3–C3: Se*_y_*(Ge_0.1_As_0.3_Te_0.6_)_100−*y*_ (*y* = 5, 10, 14) glasses is shown in Figure 6a,b, respectively. With the decrease of the Te content, the refractive index of the A2–A6 glasses decreases gradually. Similarly, the refractive index of the A3, B3, and C3 glasses decreases with the increase of the Se content. Generally, the refractive index of glass is related to density and ionic polarization. When the atoms in the glass are packed more closely together, the density of the glass increases. As shown in Table 1, when the Se content is fixed, the molar volume and density of the A2–A6 glasses decrease with the decrease of the Te content, which means that the glass with the high Te content has closer atomic packing and a higher density, so it has a higher refractive index. Since the atomic radius and mass of the Se atom are both smaller than that of the Te atom, the density of the A3–C3 glasses decreases with the increase of the Se content, thus presenting a decreasing refractive index of the glass.

### 3.3. Structure Analysis

To evaluate the structural modification caused by the introduction of the Se and Te element, Raman spectra of A2–C2: Se*_x_*(Ge_0.1_As_0.2_Te_0.7_)_100−*x*_ (*x* = 5, 10, 14) and B1–B6: Se_10_(Ge_0.1_As_0.9−*y*_Te*_y_*)_90_ (*y* = 0.8 − 0.3) glasses are shown in Figure 7a,b, respectively. Two main vibration bands at 116 and 136 cm^−1^ are observed in GAST glasses. With the increase of the Se content, the vibration band at 88 cm^−1^ disappeared in the Se*_x_*(Ge_0.1_As_0.2_Te_0.7_)_100−*x*_ glasses, and the intensity of the shoulder vibration band at 166 cm^−1^ gradually increases. In GAST glasses, the bonding energies of the possible bonds decreased in the following order: Ge-Se (49.10 kcal/mol) > As-Se (41.69 kcal/mol) > Ge-Te (35.50 kcal/mol) > Te-Te (33.30 kcal/mol) > As-Te (32.70 kcal/mol) [22]. The Ge-Se bond will form first and its number will also increase with the increase of the Se content. However, the Se*_x_*(Ge_0.1_As_0.2_Te_0.7_)_100−*x*_ glasses are rich in Te, with a low Se content and no characteristic vibration peak of GeSe_2_ at 200 cm^−1^. Thus, Se atoms will form the [GeTe_4_Se_4−*x*_] tetrahedra by breaking the GeTe_4_ tetrahedra, resulting in the vibration band at 136 cm^−1^ [23]. The vibration band at 116 cm^−1^ should be attributed to the symmetric stretching vibrations of the Ge-Te tetrahedra and the symmetric bending vibration of the As-Te triangular cone [17,24]. The vibration peak at 88 cm^−1^ disappears with the increase of the Se content, which should be attributed to the anti-symmetric stretching vibration of the Te_3_ triangular cone, and the band at 166 cm^−1^ is assigned to the anti-symmetric bending vibration of AsTe_3_ [22].

When the Se content is fixed, two vibration bands at 116 and 136 cm^−1^ of the Se_10_(Ge_0.1_As_0.9−*y*_Te*_y_*)_90_ glasses gradually move to a high frequency, and merge into a broad peak package with the decrease of the Te content. To better understand the structure evolution, Raman spectra of Se_10_(Ge_0.1_As_0.9−*y*_Te*_y_*)_90_ glasses were deconvoluted by Gaussion multi-peak fitting. As shown in Figure 8, when Te ≥ 50 mol %, the vibration bands of glass are mainly composed of the symmetric stretching of GeTe_4_, symmetric bending of AsTe_3_, and a [GeTe_4_Se_4−*x*_] stretching vibration, which is corresponding to the bands at 116 cm^−1^, 118 cm^−1^, and 136 cm^−1^, respectively. The vibration band at 155 cm^−1^ is assigned to the symmetric stretching of the Te_3_ pyramid. While Te ≤ 50 mol%, there are not enough Te atoms for bonding in the Te-deficient glass, thus the vibration bands of 168 and 188 cm^−1^ should be attributed to As-As and Ge-Ge homopolar bonds, respectively [25].

### 3.4. AC Electrical Conductivity

The AC conductivity σ of glass can be obtained by the following formula,
(1)$$\sigma_{\mathrm{AC}}=\frac{t}{AR_{S}}$$
where *t* and *A* are the thickness and surface area of samples, respectively. The resistance value *R_s_* of the sample can be obtained through the AC impedance plots. For example, the A11 sample is investigated at temperatures between 293 and 333 K (Figure 9). *Z*′ and *Z*″ are the real and imaginary parts of the impedance, respectively. The resistance of the sample can be obtained by fitting the semicircle diameter of the curve. It is clear that there are no polarization arms on the impedance graph, indicating the conductive type of the glass is electronic-type conductivity. Besides this, it is not difficult to find that the resistance of the glass sample decreases with increasing the temperature, indicating the conductivity of the sample increases. In fact, the AC conductivity of glass varies with temperature in accordance with the Arrhenius equation,
(2)$$\sigma_{\mathrm{AC}}=\frac{\sigma_{0}}{T}\exp(-\frac{E_{a}}{kT})$$
where *σ*_0_ is the pre-exponential factor, *E*_a_ is the activation energy, *k* is Boltzmann constant, and *T* is the Kelvin temperature, respectively. According to Figure 10, the electrical conductivity values conform reasonably well to the Arrhenius law at the range of the measurement temperature. The *σ* value of the investigated glass increases with the increase of the temperature. The influence of temperature on electronic conductivity mainly includes the carrier mobility and carrier concentration, and the latter is the main reason in this work. It is common in electron (or hole) semiconductors and ionic conductors, because these two conductors are generally viewed as a thermal-activation process that jumps over the energy barrier [11,26,27,28]. That is to say, as the temperature increases, the concentration of carriers per unit volume inside the glass increases, and so does its conductivity. In addition, *E*_a_ and *σ*_0_ can be obtained by fitting the slope and intercept of a straight line, respectively. The values of *σ*, *δ*, *E*_a_, and *σ*_0_ are tabulated in Table 2. When the Se content is fixed, Te presents a positive effect on the *σ*, and the glass exhibits a higher *σ* value with the increase of the Te content. On the contrary, when the Ge-As-Te ternary composition is fixed, Se plays a negative role on the *σ*, and the *σ* values of the glasses decrease with the increase of Se content. In the studied glasses, the Se_5_(Ge_0.1_As_0.2_Te_0.7_)_95_ glass has the best *σ* value (1.55 × 10^−5^ S/cm), which is relatively high for the semiconductor glass. In addition, the activation energy *E*_a_ can be regarded as a parameter reflecting the sensitivity of the reaction rate to temperature [29,30]. For the A7–C7 glasses, the *E*_a_ values of the glasses increase with the increase of the Se content, because Se is generally less conductive than Te, which means that the glass with a higher *E*_a_ value has a lower *σ* value. In the same way, for the A2–A6 or A7–A11 glasses, the *E*_a_ values should decrease with the increase of the Te content because the high Te composition of the glass has a greater *σ* value. In fact, all the analyses mentioned above are illustrated in Table 2. On the other hand, it is difficult to interpret the theoretical meaning of the pre-exponential factor *σ*_0_. Some researchers suggested that it is correlated with the number of mobile charges, while others considered *σ*_0_ as independent of compositions [30,31]. Therefore, only relevant parameter data are collected in this work, and its theoretical significance is not discussed.

## 4. Conclusions

In summary, Ge-As-Se-Te glasses were prepared by the melt quenching technique. The thermophysical, optical, and electrical properties of GAST glass were studied. The *T*_g_ and *T*_x_ values of the glasses increase with the decrease of the Te content, and the introduction of Se improves the anti-crystallization of the Ge-As-Te glass. Some glasses with Δ*T* ≥ 150 °C have an excellent thermal stability, indicating these glasses can be prepared with a large size for industrialization. The GAST glass still has a wide IR transmission window (3–18 μm) and high linear refractive index (3.2–3.6), indicating the glass can be used for IR optics. Raman spectra show that the main structural units for the GAST glass are [GeTe_4_] tetrahedra, [AsTe_3_] pyramids, and [GeTe_4_Se_4−*x*_] tetrahedra; As-As and Ge-Ge homopolar bonds appear in the glass when the Te content is less than 50 mol%. The conductivity *σ* of the studied GAST glasses decrease with the decrease of the Te content. The highest *σ* value of 1.55 × 10^−5^ S/cm is obtained in the glass with a high Te content. Besides this, the *σ* value of the glass increases with temperature, which can be interpreted as the thermal activation of electrons jumping over the energy barrier. The activation energy *E*_a_ of the glass increases with the decrease of the Te content, indicating that glass with a high Te content is more sensitive to temperature. This work provides a basis for expanding the application of GAST glass materials in the field of IR optical devices.

## Figures and Tables

**Figure 1 materials-15-01797-f001:**
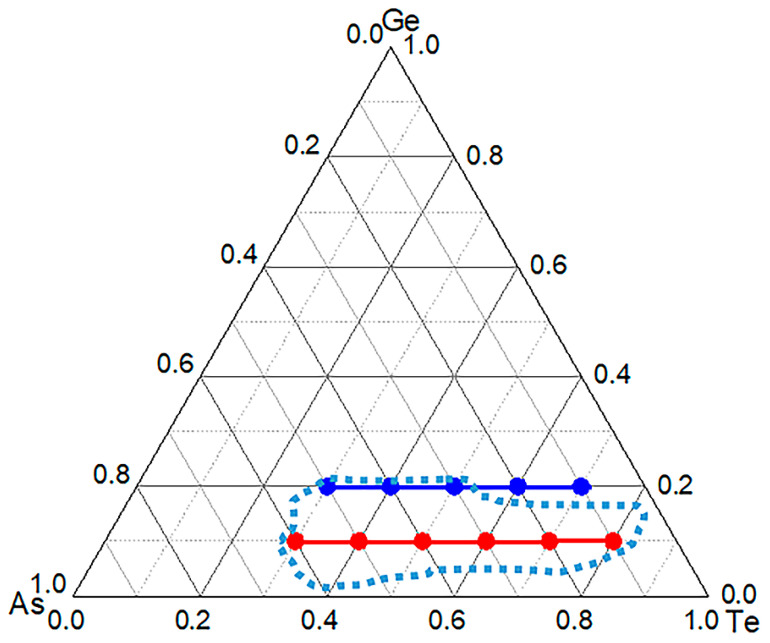
Glass-forming region of the Ge-As-Te glasses and two lines selected in this work (red and blue points).

**Figure 2 materials-15-01797-f002:**
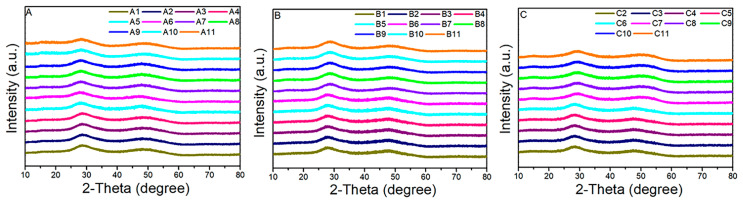
XRD diffraction patterns of GAST glasses. (**A**): A1–A6: Se_5_(Ge_0.1_As_0.9−*x*_Te*_x_*)_95_, (*x* = 0.8 − 0.3), A7–A11: Se_5_(Ge_0.2_As_0.8−*x*_Te*_x_*)_95_, (*x* = 0.7 − 0.3); (**B**): B1-B6: Se_10_(Ge_0.1_As_0.9−*x*_Te*_x_*)_90_, (*x* = 0.8 − 0.3), B7–B11: Se_10_(Ge_0.2_As_0.8−*x*_Te*_x_*)_90_, (*x* = 0.7 − 0.3); (**C**): C2–C6: Se_14_(Ge_0.1_As_0.9−*x*_Te*_x_*)_86_, (*x* = 0.7 − 0.3), C7–C11: Se_14_(Ge_0.2_As_0.8−*x*_Te*_x_*)_86_, (*x* = 0.7 − 0.3).

**Figure 3 materials-15-01797-f003:**
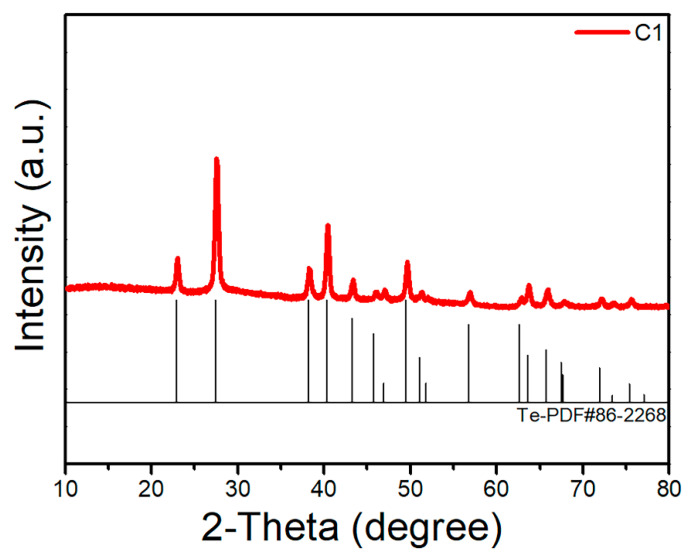
XRD diffraction patterns of C1 sample.

**Figure 4 materials-15-01797-f004:**
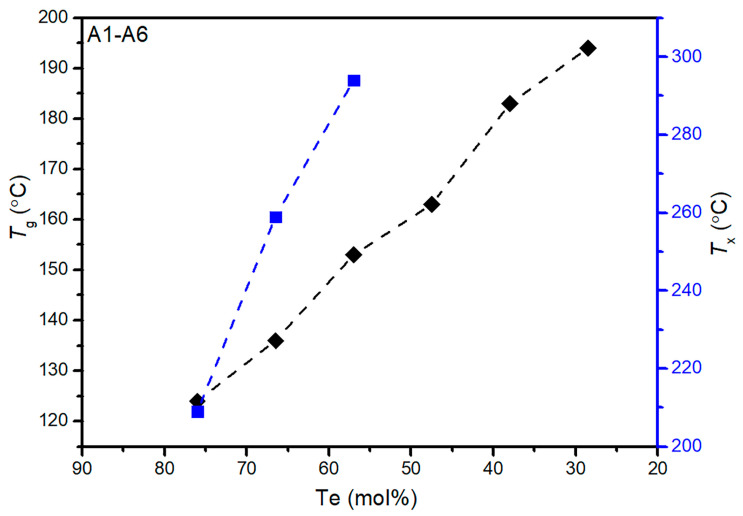
*T_g_* and *T_x_* values of A1–A6: Se_5_(Ge_0.1_As_0.9−*x*_Te*_x_*)_95_ (*x* = 0.8, 0.7, 0.6, 0.5, 0.4, 0.3) glasses.

**Figure 5 materials-15-01797-f005:**
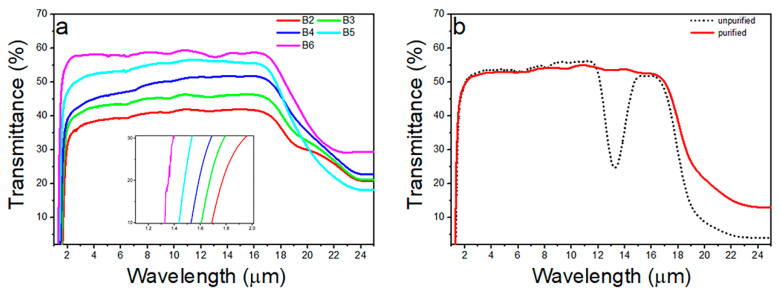
(**a**) Transmission spectra of B2–B6: Se_10_(Ge_0.1_As_0.9−*x*_Te*_x_*)_90_ (*x* = 0.7 − 0.3) glasses (inset is enlarge of the short-wave cut-off edge) and (**b**) transmission spectra of B11: Se_10_(Ge_0.2_As_0.5_Te_0.3_)_90_ glass before and after purification.

**Figure 6 materials-15-01797-f006:**
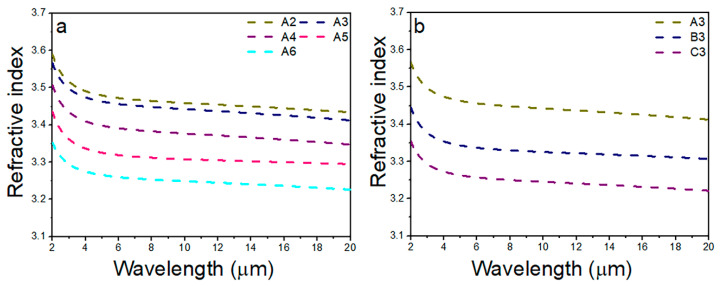
The linear refractive index of GAST glasses: (**a**) A2–A6: Se_5_(Ge_0.1_As_0.9−*x*_Te*_x_*)_95_ (*x* = 0.7 − 0.3), (**b**) A3–C3: Se*_y_*(Ge_0.1_As_0.3_Te_0.6_)_100−*y*_ (*y* = 5, 10, 14).

**Figure 7 materials-15-01797-f007:**
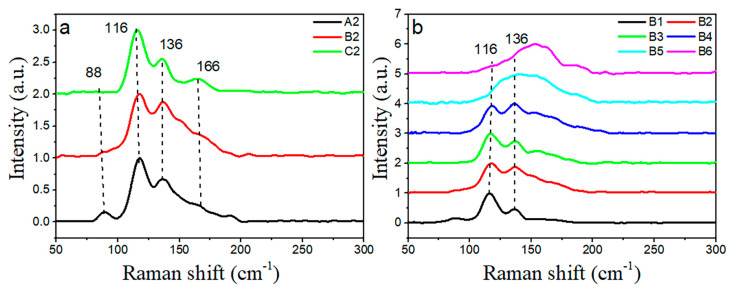
Raman spectra of glasses: (**a**) A2–C2: Se*_x_*(Ge_0.1_As_0.2_Te_0.7_)_100−x_ (*x* = 5, 10, 14) and (**b**) B1–B6: Se_10_(Ge_0.1_As_0.9−*y*_Te*_y_*)_90_ (*y* = 0.8, 0.7, 0.6, 0.5, 0.4, 0.3), respectively.

**Figure 8 materials-15-01797-f008:**
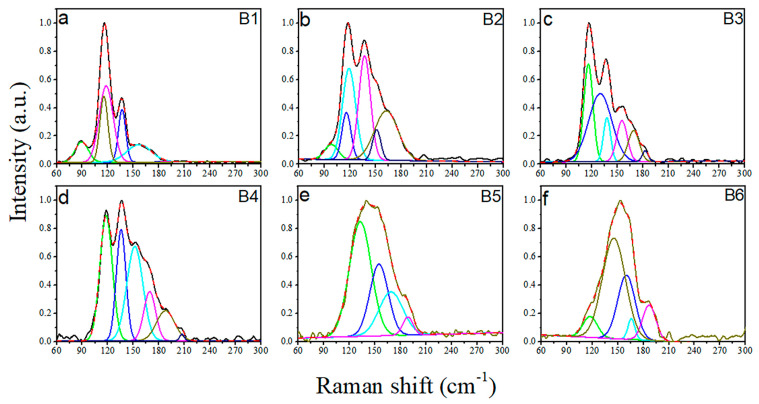
Deconvolution of Raman spectra of B1–B6: Se_10_(Ge_0.1_As_0.9−*y*_Te*_y_*)_90_ glasses: (**a**) B1: Se_10_(Ge_0.1_As_0.1_Te_0.8_)_90_, (**b**) B2: Se_10_(Ge_0.1_As_0.2_Te_0.7_)_90_, (**c**) B3: Se_10_(Ge_0.1_As_0.3_Te_0.6_)_90_, (**d**) B4: Se_10_(Ge_0.1_As_0.4_Te_0.5_)_90_, (**e**) B5: Se_10_(Ge_0.1_As_0.5_Te_0.4_)_90_, and (**f**) B6: Se_10_(Ge_0.1_As_0.6_Te_0.3_)_90_, respectively.

**Figure 9 materials-15-01797-f009:**
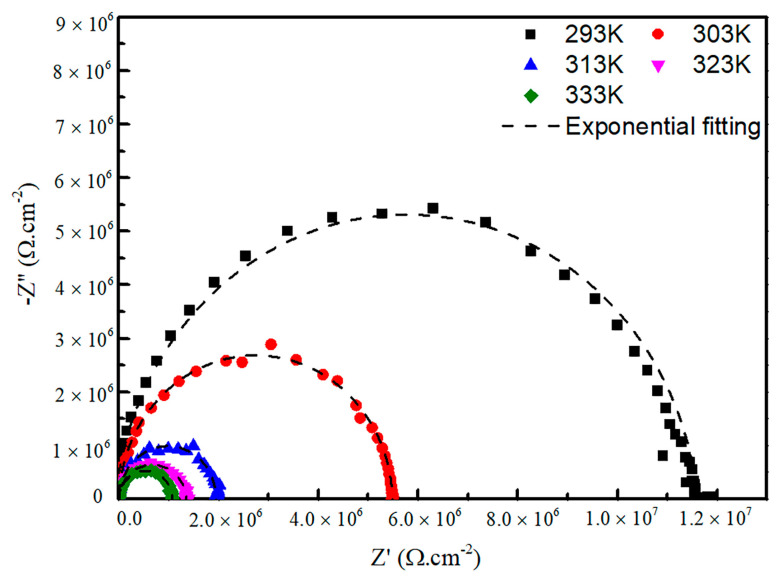
AC impedance of A11 glass at different temperatures.

**Figure 10 materials-15-01797-f010:**
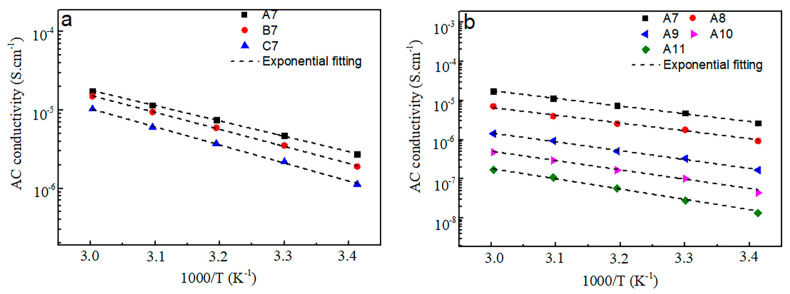
Temperature dependence on the AC electrical conductivities of GAST glasses: (**a**) Se*_x_*(Ge_0.2_As_0.1_Te_0.7_)_100−*x*_ (*x* = 5, 10, 14) and (**b**) Se_5_(Ge_0.2_As_0.8−*y*_Te*_y_*)_95_ (*y* = 0.7, 0.6, 0.5, 0.4, 0.3).

**Table 1 materials-15-01797-t001:** DSC measurement results, mean coordination number (MCN), molar mass (*M*_w_), molar volume (*M*_v_), and density (*ρ*) of GAST glasses.

Number	Compositions	*T_g_* (°C)	*T_x_* (°C)	Δ*T* (°C)	MCN	*M_w_* (±0.01 g/mol)	*M_v_* (±0.01 cm^3^/mol)	*ρ* (±0.001 g/cm^3^)
A1	Se_5_(Ge_0.1_As_0.1_Te_0.8_)_95_	124	209	85	2.3	114.94	20.65	5.567
A2	Se_5_(Ge_0.1_As_0.2_Te_0.7_)_95_	136	259	123	2.4	109.93	20.00	5.496
A3	Se_5_(Ge_0.1_As_0.3_Te_0.6_)_95_	153	294	141	2.5	104.93	19.33	5.428
A4	Se_5_(Ge_0.1_As_0.4_Te_0.5_)_95_	163	-	-	2.6	99.93	18.71	5.342
A5	Se_5_(Ge_0.1_As_0.5_Te_0.4_)_95_	183	-	-	2.7	94.92	17.90	5.304
A6	Se_5_(Ge_0.1_As_0.6_Te_0.3_)_95_	194	-	-	2.7	89.92	17.07	5.268
A7	Se_5_(Ge_0.2_As_0.1_Te_0.7_)_95_	170	309	139	2.5	109.72	20.21	5.428
A8	Se_5_(Ge_0.2_As_0.2_Te_0.6_)_95_	188	335	147	2.6	104.71	19.63	5.333
A9	Se_5_(Ge_0.2_As_0.3_Te_0.5_)_95_	218	372	154	2.7	99.71	19.03	5.238
A10	Se_5_(Ge_0.2_As_0.4_Te_0.4_)_95_	243	-	-	2.8	94.70	18.21	5.201
A11	Se_5_(Ge_0.2_As_0.5_Te_0.3_)_95_	264	-	-	2.9	89.70	17.44	5.144
B1	Se_10_(Ge_0.1_As_0.1_Te_0.8_)_90_	116	185	69	2.3	113.05	20.61	5.485
B2	Se_10_(Ge_0.1_As_0.2_Te_0.7_)_90_	146	260	114	2.4	108.30	20.18	5.367
B3	Se_10_(Ge_0.1_As_0.3_Te_0.6_)_90_	153	298	145	2.5	103.56	19.45	5.323
B4	Se_10_(Ge_0.1_As_0.4_Te_0.5_)_90_	165	-	-	2.5	98.82	18.92	5.224
B5	Se_10_(Ge_0.1_As_0.5_Te_0.4_)_90_	194	-	-	2.6	94.08	18.32	5.136
B6	Se_10_(Ge_0.1_As_0.6_Te_0.3_)_90_	213	-	-	2.7	89.34	17.48	5.110
B7	Se_10_(Ge_0.2_As_0.1_Te_0.7_)_90_	161	-	-	2.5	108.10	20.27	5.334
B8	Se_10_(Ge_0.2_As_0.2_Te_0.6_)_90_	180	-	-	2.5	103.36	19.79	5.223
B9	Se_10_(Ge_0.2_As_0.3_Te_0.5_)_90_	207	-	-	2.7	98.61	18.97	5.199
B10	Se_10_(Ge_0.2_As_0.4_Te_0.4_)_90_	231	-	-	2.8	93.87	18.33	5.122
B11	Se_10_(Ge_0.2_As_0.5_Te_0.3_)_90_	278	-	-	2.9	89.13	17.55	5.079
C1	Se_14_(Ge_0.1_As_0.1_Te_0.8_)_86_	-	-	-	2.3	111.53	20.60	5.413
C2	Se_14_(Ge_0.1_As_0.2_Te_0.7_)_86_	117	243	126	2.3	107.00	19.97	5.359
C3	Se_14_(Ge_0.1_As_0.3_Te_0.6_)_86_	139	290	151	2.4	102.47	19.48	5.259
C4	Se_14_(Ge_0.1_As_0.4_Te_0.5_)_86_	158	-	-	2.5	97.94	18.82	5.203
C5	Se_14_(Ge_0.1_As_0.5_Te_0.4_)_86_	193	-	-	2.6	93.41	18.30	5.104
C6	Se_14_(Ge_0.1_As_0.6_Te_0.3_)_86_	221	-	-	2.7	88.88	17.57	5.058
C7	Se_14_(Ge_0.2_As_0.1_Te_0.7_)_86_	156	-	-	2.4	106.80	20.23	5.280
C8	Se_14_(Ge_0.2_As_0.2_Te_0.6_)_86_	180	-	-	2.5	102.27	19.67	5.199
C9	Se_14_(Ge_0.2_As_0.3_Te_0.5_)_86_	194	-	-	2.6	97.74	19.02	5.138
C10	Se_14_(Ge_0.2_As_0.4_Te_0.4_)_86_	230	-	-	2.7	93.21	18.55	5.026
C11	Se_14_(Ge_0.2_As_0.5_Te_0.3_)_86_	261	-	-	2.8	88.68	17.74	4.999

“-” does not appear in the test curve.

**Table 2 materials-15-01797-t002:** Electrical conductivities *σ*, resistivity *δ*, activation energy *E*_a_, pre-exponential factor *σ*_0_ of A2–A11, B7, and C7 glasses (T = 293 K).

Number	Compositions	*σ* (S·cm^−1^)	*δ* (Ω·cm)	*E*_a_ (eV)	Log_10_ σ_0_ (S·cm^−1^)
A2	Se_5_(Ge_0.1_As_0.2_Te_0.7_)_95_	1.55 × 10^−5^	6.46 × 10^4^	0.147	6.69
A3	Se_5_(Ge_0.1_As_0.3_Te_0.6_)_95_	7.57 × 10^−6^	1.32 × 10^5^	0.169	7.00
A4	Se_5_(Ge_0.1_As_0.4_Te_0.5_)_95_	1.96 × 10^−6^	5.09 × 10^5^	0.180	7.08
A5	Se_5_(Ge_0.1_As_0.5_Te_0.4_)_95_	1.04 × 10^−6^	9.61 × 10^5^	0.203	7.58
A6	Se_5_(Ge_0.1_As_0.6_Te_0.3_)_95_	3.00 × 10^−7^	3.33 × 10^6^	0.214	7.41
A7	Se_5_(Ge_0.2_As_0.1_Te_0.7_)_95_	2.16 × 10^−6^	4.63 × 10^5^	0.170	6.85
A8	Se_5_(Ge_0.2_As_0.2_Te_0.6_)_95_	9.35 × 10^−7^	1.07 × 10^6^	0.172	6.48
A9	Se_5_(Ge_0.2_As_0.3_Te_0.5_)_95_	1.69 × 10^−7^	5.91 × 10^6^	0.195	6.63
A10	Se_5_(Ge_0.2_As_0.4_Te_0.4_)_95_	4.44 × 10^−8^	2.25 × 10^7^	0.205	6.52
A11	Se_5_(Ge_0.2_As_0.5_Te_0.3_)_95_	1.35 × 10^−8^	7.38 × 10^7^	0.226	6.82
B7	Se_10_(Ge_0.2_As_0.1_Te_0.7_)_90_	1.31 × 10^−6^	7.61 × 10^5^	0.187	7.38
C7	Se_14_(Ge_0.2_As_0.1_Te_0.7_)_86_	1.13 × 10^−6^	8.86 × 10^5^	0.199	7.62

## Data Availability

The raw data required to reproduce these results cannot be shared at this time as the data also form part of an ongoing study.

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
