# Peer review of "Effect of Se on Structure and Electrical Properties of Ge-As-Te Glass"

_materials, 2022, doi:10.3390/ma15051797_

Round 1
Reviewer 1 Report
Comments on MDPI_GAST
The manuscript titled “Effect of Se on structure and electrical properties of Ge-As-Te glasses” by Liu et al., has been reviewed and the comments are given below.
Authors have synthesised Se doped Ge-As-Te glasses and studied the effect of selenium on structural and electrical properties of Ge-As-Te glasses. Though a systematic study has been done on large number of compositions, it is necessary for the authors to explain few important points in their work.
- Title indicates the effect of Se on structure & electrical properties, while the introduction mainly deals with the optical properties. Selenium in general is less conducting compared to tellurium and hence it will be better to quote some literature on electrical properties of selenium & tellurium doped glasses in the introduction section.
- In continuation, discussion on electrical properties is not enough to substantiate the title of the paper. More detailed discussion on the effect of selenium in modifying the electrical conductivity of GAST glasses in needed.
- References are too old. Check for recent ones as there are hardly 5 references after 2016. Also to add more suitable references.
- English corrections in few places – Line number-127-130; Line number 137 (even crystallization??); 171-172; 188 – Vibration; 207 – Deconvoluted; Line numbers – 212-213; Line number – 246;
- Title of Table 2 to start in Capitals
Authors are required to modify as indicated above and do language corrections before the article can be accepted for publication.
Reviewer 2 Report
Reviewer Comments:
In this paper author aim to introduced Se to stabilize the Ge-As-Te glass and author also investigated the thermo-physical, optical and electrical properties of the Ge-As-Se-Te (GAST) glasses were studied. The finding of this paper is interesting and author also characterize the material by using FTIR, DSC, and XRD, however author need to elaborate more especially the introduction part with more reference. More related discussion should be included in the introduction section. The results are interesting and therefore, I would like to recommend this manuscript in the journal of Materials after these minor corrections.
Comments:
- Se is already used by researcher in GAST glass, Preparation of Ge-As-Se-Te(GAST) glasses is not new at different ratio, Please highlight the novelty of your work.
- Why the amount of Ge-As-Te is very less as compare to Se in most of the compositions?
- Why AC conductivity of the GAST glass decreases with increasing temperature?
- Why author selected to introduce different ratio of Se (5, 10, 14 mol% Se)?
- Which ratio provide best result in term of thermo-physical, optical and electrical properties? Author should provide best ration in abstract and conclusion.
Reviewer 3 Report
The paper “Effect of Se on structure and electrical properties of Ge-As-Te glass” is devoted to preparation and investigation of the Ge-As-Se-Te glass systems. The thermophysical, optical and electrical properties of the prepared glass samples were studied. DSC, XRD, FTIR, ellipsometry, Raman spectroscopy techniques were applied for the samples investigation. The topic of this paper is critically actual especially for the areas of infrared optics. The data are reliable and do not cause much doubt. Nevertheless, there are several points before the paper can be published. I hope that authors after major revisions can improve the paper and can publish it in Materials.
- The novelty of the research must be highlighted in Introduction part.
- The Introduction part must be improved with literature in the field of functional glass systems and I suggest to use the following reference (see and discuss: doi:10.1117/12.956906; doi:10.3390/ma14143772).
- The description of the experimental samples should be added to the Materials and Methods part. Now it is difficult understand how many samples? What the difference between them?
- Figure 2 – what color does correspond to which sample? Please include this information in Figure legend or caption.
- How do you explain the XRD behavior for the C1 sample? Why does the diffractogram differ from other samples?
- Manuscript contains the DSC data about 33 samples, but the other experimental techniques are used for the some series of samples. Why so?
- The practical recommendation should be added to the text.
- The Conclusion part is too short, please improve it.
- The list of References is formatted not in the MDPI Materials requirements. Please use the necessary citation style.
- There are some insufficient typos and English mistakes in the text.
But any way I impressed by this paper. But authors must explain some details and improve the paper in accordance with my comments. The paper should be sent to me for the second analysis after the major revisions.
Round 2
Reviewer 1 Report
Minor English Language corrections needed to remove grammatical errors.
Author Response
Thanks the reviewer's comments. we have carefully checked the manuscript again and made corresponding grammatical corrections.
Reviewer 3 Report
The paper can be puplished in present form.Author Response
Thanks a lot! We have carefully checked the English again and made corresponding corrections.